# Questionnaire on the Current Status and Awareness of Palliative Medicine by Community Cooperation Pharmacies

**DOI:** 10.3390/pharmacy12040110

**Published:** 2024-07-16

**Authors:** Keigo Nagatani, Mayu Nakahara, Sachiko Omotani, Michiaki Myotoku

**Affiliations:** 1Faculty of Pharmacy, Osaka Ohtani University, 3-11-1 Nishikiori-kita, Osaka 584-8540, Japan; kei50nagatani@gmail.com (K.N.);; 2Yao Municipal Hospital, 1-3-1 Ryuge-cho, Osaka 581-0069, Japan

**Keywords:** community cooperation pharmacies, palliative medicine, medical narcotics, aseptic preparation facility, transfer of medical narcotics

## Abstract

Community cooperation pharmacies are equipped to prepare narcotics and sterile injectable drugs for palliative medicine at home for cancer pain and end-of-life care; however, to the best of our knowledge, the actual status of the system to provide palliative medicine at home has not yet been examined. Therefore, in this study, given that home palliative medicine is one of the accreditation criteria for community cooperation pharmacies, a questionnaire survey was conducted among managing pharmacists engaged in community cooperation pharmacies to investigate the actual status of the system to provide appropriate services, mainly pain management, to patients who need home palliative medicine. An analysis of responses to the questionnaire showed that pharmacists working in community cooperation pharmacies had a high level of understanding of the proper use of rescue doses of medical narcotics and patient guidance. Pharmacists with experience in sterile and injection preparations also had a high level of understanding of palliative medicine. On the other hand, they had a low level of understanding of the WHO method for cancer pain treatment and appropriate suggestions for opioid switching. These results indicate that the creation of learning opportunities, such as training on injectables and prescription designs, for pharmacists in community cooperation pharmacies is one of the measures that may improve their understanding of palliative medicine.

## 1. Introduction

One of the roles of the pharmacist is to provide drug therapy in collaboration with multiple professions so that patients may continue to remain at home until the very end. Therefore, to realize and enhance the seamless transition from hospitalization to palliative medicine at home, collaborations between pharmacies and hospital/clinic pharmacists need to be strengthened. In Japan, pharmacists working in pharmacies and hospitals involved in palliative medicine are primarily responsible for checking the condition of patients undergoing cancer treatment and other therapies, and proposing and implementing drug therapies, mainly medical narcotics, according to individual symptoms and conditions. They also play a role in supporting doctors and patients from the medical side, for example, by providing pharmacological support in the event of side effects.

In Japan, according to the “Act on Quality, Efficacy and Safety Assurance of Pharmaceuticals, Medical Devices and Other Products” enacted in November 2019, in August 2021, a certification system for community cooperation pharmacies and pharmacies cooperating with medical institutions specializing in specific functions was launched to enable patients to choose the pharmacy that best suits their needs [1]. According to the Ministry of Health, Labor and Welfare Ordinance (2021) [1], regarding home palliative medicine, the standards stipulate ‘the development of a system for receiving and dispensing narcotics’, ‘the development of a system for implementing aseptic dispensing procedures’, and ‘the implementation of initiatives related to home healthcare’ for dispensing and sales systems to ensure a stable supply of medicines to local users and dispensing and guidance systems at home and other places. In a study by Hasegawa et al. [2], it was reported that pharmacies certified as community cooperation pharmacies were more involved in dispensing medical narcotics and end-of-life care than pharmacies that were not certified, especially in providing home care and 24 h service to patients in the community. Therefore, although it is clear from accreditation criteria that community cooperation pharmacies have a central role, at least with regard to home palliative medicine, there has been no survey of their actual status.

The present study conducted a questionnaire for managing pharmacists engaged in community cooperation pharmacies to investigate the actual status of the system to provide appropriate service, mainly pain management, to patients requiring home palliative medicine given that this is an accreditation criterion.

Previous studies by Myotoku et al. [3], Harikae et al. [4], and Hidaka et al. [5] surveyed pharmacists working in various pharmacies before the start of the community cooperation pharmacies accreditation system in 2021, so the level of pharmacies was unknown. The results reported that pharmacists working in pharmacies that provide medical drug handling and palliative medicine at home had a higher level of understanding and knowledge proficiency regarding medical drugs and palliative medicine at home than pharmacists working in pharmacies that do not.

In this study, a questionnaire was developed based on these findings [3,4,5], and the level of understanding of pharmacists working in community cooperation pharmacies regarding ‘pain assessment using various scales’, ‘WHO method of cancer pain treatment’, ‘appropriate use of rescue doses and patient guidance’, and ‘appropriate consultation for opioid switching’ was surveyed and evaluated.

## 2. Materials and Methods

### 2.1. Target Facilities and Survey Method and Duration

Pharmacists managing 563 community partnership pharmacies listed on the administrative website of the Kinki region (Osaka, Kyoto, Hyogo, Nara, Wakayama, and Shiga prefectures) as of April 2023, were targeted. Questionnaires were distributed by post. Responses were collected in an unsigned self-addressed envelope or via the web. When completing the questionnaire, respondents were given a written explanation of the significance and purpose of the study, the expected disadvantages of participating, and that participation was voluntary and did not contain personal information. Participation in the questionnaire was deemed to constitute consent to participate in the study. The survey period was set for 7 June–31 July 2023.

### 2.2. Questionnaire Items

For the survey items of the questionnaire, factor items were selected with reference to previous studies [3,4,5], and their validity and appropriateness were examined through discussions with four pharmacists working in community cooperation pharmacies who are familiar with palliative medicine at home, and are shown in Table 1.

The questionnaire included the following four items: ‘Facilities and personnel in charge (item I)’, ‘Current status of home healthcare and drug-drug collaborations (item II)’, ‘Sterile equipment and preparation of injections and infusions in pharmacies (item III)’, and ‘Guidance and handling of narcotic drugs (item IV)’, and one item, ‘Proper use of medical narcotics and palliative drug therapy (item V)’, was used to assess the level of understanding. Responses to item V were asked on a four-point scale.

### 2.3. Evaluation Items

Responses to items in collected questionnaires were tabulated. Regarding item V, Q18, “Pain assessment using various scales”, Q19, “WHO method of cancer pain treatment”, Q20, “Appropriate use of rescue doses and patient guidance”, and Q21, “Appropriate consultation for opioid switching”, the percentage of respondents who answered ‘know in detail’ or ‘know to some extent’ was calculated as the 2Top Ratio, which was evaluated as the percentage of the highly rated group for understanding [6]. In addition, “Have heard of it” and “Don’t know” were compared as the 2Under Ratio.

To identify the factors that led to a higher level of understanding, item I Q2, “Years of experience as a pharmacy pharmacist”, item I Q3, “Experienced Pharmaceutical-Related Work Experience”, item I Q6, “What type of accreditation does the pharmacy you work for have?”, item II Q7, “Experience in dealing with end-of-life palliative care patients with cancer who are highly dependent on medical care”, item III Q11, “Experience in sterile preparation”, and item IV Q16, “Dispensing experience of medical narcotics by dosage form”, were used as indicators. The results of the questionnaire were cross-tabulated with responses to Q18–21 in item V. The 2Top Ratio was calculated for each item.

### 2.4. Statistical Analysis

Fisher’s exact probability test and Pearson’s chi-square test were used to test the significance of differences in response rates by category to the question, and the significance level was set at 0.05. For those significantly different by Pearson’s chi-square test, Spearman’s rank correlation coefficient, a nonparametric measure, was calculated.

The survey items for this study were developed based on previous studies [3,4,5] and discussions with four pharmacists working in community cooperation pharmacies who are familiar with palliative medicine at home. Therefore, Cronbach’s coefficient (α) was used to confirm the internal consistency of the response data for each item. Question items were rated as reliable if they were 0.7 or higher.

IBM SPSS Statistics Desktop Ver. 21 (IBM Japan, Tokyo, Japan) was used for statistical analyses.

### 2.5. Ethical Considerations

The present study was conducted in compliance with the “Ethical Guidelines for Medical and Biological Research Involving Human Subjects” and with the approval of the Bioethics Committee of the Faculty of Pharmaceutical Sciences, Osaka Ohtani University (BE-0080-23), approved on 2 June 2023.

## 3. Results

### 3.1. Survey, Facilities, and Persons in Charge

The questionnaire collection rate was 47.2% (266 facilities). Cronbach’s coefficient (α) was 0.90 for “Palliative drug therapy and proper use of medical narcotics”. Cronbach’s coefficients (α) for each item were 0.82 for “years of experience as a pharmacy pharmacist”, 0.82 for “experienced pharmaceutical-related work experience”, 0.84 for “what type of accreditation does the pharmacy you work for have?”, 0.85 for “experience in dealing with end-of-life palliative care patients with cancer who are highly dependent on medical care”, 0.85 for “experience in sterile preparation”, and 0.84 for “dispensing experience of medical narcotics by dosage form”. The results showed a certain degree of reliability.

The demographics of the respondents are shown in Table 2. In terms of age, 80 respondents (30.1%) were in their 30s and 80 (30.1%) were in their 40s, accounting for 60.2% of the total. In terms of years of experience, 77 respondents (28.9%) were pharmacy pharmacists with more than 21 years of experience, 160 (60.2%) had only experience as pharmacy pharmacists, and the average number of pharmacists in community cooperation pharmacies was 5.6 ± 3.4.

A total of 135 pharmacies (50.8%) were accredited only as community cooperation pharmacies, 119 (44.7%) had two accreditations as community cooperation pharmacies and health support pharmacies), 5 (1.9%) had two accreditations as community cooperation pharmacies and specialized medical institution cooperation pharmacies, and 7 (2.6%) had three accreditations: community cooperation pharmacies, specialized medical institution cooperation pharmacies, and health support pharmacies.

### 3.2. The Current State of Home Care and Drug–Drug Collaborations

A total of 130 pharmacists (48.9%) had experience in dealing with end-of-life palliative care patients with cancer and high medical dependency (e.g., patients receiving high-calorie infusions or injectable medical narcotics). In addition, 116 pharmacists (43.6%) had participated (including via the web) in joint guidance at discharge (pre-discharge conferences) held in hospitals.

Overall, 64.6% of pharmacists dealing with end-of-life palliative care patients with cancer and high medical dependency had attended pre-discharge conferences, whereas 35.4% did not (Table 3).

A total of 253 (95.1%) pharmacists had attended conferences with other professions about team care at home.

### 3.3. Aseptic Facilities and Preparation of Injections and Infusions in Pharmacies

Of the 266 pharmacies that responded to this questionnaire, 62 (23.3%) had aseptic preparation facilities, whereas 148 (55.6%) did not but had a system for shared use (pharmacists engaged in dispensing in pharmacies without aseptic preparation rooms may use aseptic preparation rooms in pharmacies with other aseptic preparation rooms for aseptic preparation processing) (Table 4). In total, 66 pharmacies (24.8%) had performed aseptic preparation (including shared use), of which 19 (7.1%) had experience of aseptic preparation through shared use.

In total, 140 (52.6%) respondents ‘know in detail’ or ‘know to some extent’ about patient-controlled analgesia (PCA) pumps, while 126 (47.3%) had ‘heard of it’ or ‘did not know’ of it.

In response to the question “What types of PCA pumps are you familiar with?”, 161 (60.5%) pharmacists answered disposable (e.g., Balloonjector^®^), 119 (44.7%) mechanicals (e.g., CADD Series^®^), and 44 (16.5%) hybrid (COOPDECH Amy^®^). Thirty-two (12%) pharmacies owned PCA pumps and had a system for renting them to medical institutions (multiple responses).

### 3.4. Guidance and Handling of Narcotic Drugs

A total of 259 pharmacists (97.4%) had experience in dispensing prescription narcotics: 256 (96.2%) dispensed oral narcotics and 247 (92.9%) dispensed patch formulations, while 181 (68%) dispensed suppositories and 160 (60.2%) dispensed sublingual and buccal forms. In total, 76 pharmacists (28.6%) had experience with dispensing injectable drugs (Table 5), 57 pharmacists (21.4%) dispensed only ‘oral or patch formulations’, and 32 (12.0%) had dispensed both ‘oral or patch formulations and suppositories’. Ninety-four pharmacists (35.3%) had dispensed ‘only two types of oral or patch formulations and sublingual or buccal formulations’ or ‘three types of oral or patch formulations, suppositories, and sublingual or buccal formulations’.

Regarding the inter-pharmacy transfer of medical narcotics, 120 pharmacies (45.1%) obtained a permit for the inter-retailer transfer of narcotics with other pharmacies and made transfers, 37 (13.9%) obtained a transfer license but had no transfer record, 36 (13.5%) did not have a permit but would obtain one if necessary, 56 (21.1%) had no plans to obtain a permit, and 17 (6.4%) did not know.

### 3.5. Proper Use of Medical Narcotics and Palliative Drug Therapy

#### 3.5.1. Comprehension by Simple Tabulation

Regarding Q18, “Assessment of pain using various scales (numerical assessment of pain, such as NRS)”, 32 (12.0%) responded ‘know in detail’, 156 (58.7%) responded ‘know to some extent’, 70 (26.3%) responded ‘have heard of it’, and 8 (3.0%) responded ‘do not know’. The 2Top Ratio of respondents who answered ‘know in detail’ and ‘know to some extent’ was 70.7%.

Concerning Q19, “WHO method of cancer pain treatment”, 28 (10.5%) responded ‘know in detail’, 132 (49.7%) responded ‘know to some extent’, 77 (28.9%) responded ‘have heard of it’, and 29 (10.9%) responded ‘do not know’. The 2Top Ratio of respondents who answered ‘know in detail’ and ‘know to some extent’ was 60.2%.

In response to Q20, “Proper use of Rescue Dose and patient guidance”, 35 (13.2%) answered ‘know in detail’, 179 (67.3%) answered ‘know to some extent’, 43 (16.1%) answered ‘have heard of it’, and 9 (3.4%) answered ‘do not know’. The 2Top Ratio of respondents who answered ‘know in detail’ and ‘know to some extent’ was 80.5%.

Regarding Q21, “Appropriate suggestions for opioid switching”, 19 (7.1%) responded ‘know in detail’, 159 (59.8%) responded ‘know to some extent’, 68 (25.6%) responded ‘have heard of it’, and 20 (7.5%) responded ‘do not know’. The 2Top Ratio of respondents who answered ‘know in detail’ and ‘know to some extent’ was 66.9%.

#### 3.5.2. Comprehension by Cross-Tabulation

Table 6 shows the cross-tabulation results and 2Top Ratio for questions 18–21 of item V and “years of experience as a pharmacy pharmacist”. No significant differences were observed in the 2Top Ratio by years of experience as a pharmacy pharmacist for each question.

Table 7 shows the cross-tabulation results and 2Top Ratio between pharmacy pharmacists’ “experienced pharmaceutical-related work experience” and the questions in item V, Q18–21. The results of the 2Top Ratio compared to ‘only experienced pharmacy pharmacist’ and ‘experienced hospital pharmacist’ showed significant differences (*p* = 0.020, *p* = 0.023) between the categories for Q18 and Q21, but no correlation. No significant differences (*p* = 0.097, *p* = 0.338) were noted between Q19 and Q20.

Table 8 shows the cross-tabulation results and 2Top Ratio between the questions “What type of accreditation does the pharmacy you work for have?” and item V Q18–21. Regarding each question, pharmacies with only a community cooperation pharmacy accreditation had a lower 2Top Ratio than pharmacies with two accreditations, community cooperation pharmacies and health support pharmacies, as well as pharmacies with a specialized medical institution cooperation pharmacy accreditation in addition to community cooperation pharmacies or health support pharmacies. Significant differences were found between categories for all questions (*p* = 0.014, *p* = 0.030, *p* = 0.014, *p* < 0.001). Only Q21 showed a weak correlation (ρ = −0.229).

Table 9 shows the cross-tabulation results and 2Top Ratio between Q18–21 of item V and “Experience in dealing with end-of-life palliative care patients with cancer who are highly dependent on medical care”. Regarding each question, the 2Top Ratio was higher for those with a track record of responding to palliative care than for those without, with significant differences between categories for all questions (*p* < 0.001, *p* = 0.002, *p* < 0.001, *p* < 0.001).

Table 10 shows the cross-tabulation results and 2Top Ratio between Q18–21 of item V and “Experience in sterile preparation”. Regarding each question, the 2Top Ratio was higher for those with a proven track record of aseptic preparation than for those without, with significant differences between categories for all questions (*p* < 0.001, *p* = 0.009, *p* < 0.001, *p* < 0.001).

Table 11 shows the cross-tabulation results and 2Top Ratios between Q18–21 of item V and “Dispensing experience of medical narcotics by dosage form”. Regarding each question, the 2Top Ratio was significantly higher for those with a track record of dispensing injections than for those with a track record of dispensing oral or patch formulations and sublingual or buccal formulations or oral or patch formulations and suppositories and sublingual or buccal formulations, and even those with a track record of dispensing oral or patch formulations, between categories for all questions (*p* < 0.001, *p* = 0.001, *p* < 0.001, *p* < 0.001). Weak correlations (ρ = −0.398, ρ= −0.368) were found in Q18 and Q19 and moderate correlations (ρ = −0.417, ρ = −0.440) in Q20 and Q21.

## 4. Discussion

Community cooperation pharmacies are pharmacies that play a central role in palliative medicine at home, including narcotics supply. In this study, a survey was conducted to investigate the actual status of community cooperation pharmacies that provide services for patients requiring palliative medicine.

The results of this questionnaire indicated that 48.9% of pharmacists have experience in providing palliative care to patients with cancer who are highly dependent on medical care at the end of life. Among pharmacists with experience in end-of-life care, 64.6% had participated in pre-discharge conferences held by multiple disciplines, whereas 35.4% did not. According to a survey [7] by the Ministry of Health, Labor and Welfare, the number of pharmacists participating in pre-discharge conferences at pharmacies that provide home support was low at 14.8%. The results of our survey suggest that the percentage of pharmacists at community collaborative pharmacies providing end-of-life palliative care who participate in pre-discharge conferences is higher than previously reported. On the other hand, it was observed that over 50% of community collaborative pharmacies who participated in the survey were not involved in dealing with end-of-life palliative care patients with cancer and high medical dependency. While the reasons for this were not a focus of this survey, another study [8] reported that reasons for not providing this service were “no requests from physicians” and “lack of demand”. It may also be related to a lack of understanding of the willingness of pharmacists to participate in home palliative medicine and the significance of their participation by other professions [9]. Therefore, it is necessary for pharmacists in community collaborative pharmacies to actively engage in activities that raise awareness of palliative medicine among other professions and local residents and to provide information and support to home health care physicians, which will result in an increased recognition of the presence of pharmacists in home palliative medicine.

A total of 97.4% of community cooperation pharmacies had a good track record in handling narcotics. Also, only 45.1% of pharmacies reported that they had a drug retailer-to-retailer transfer license. In palliative medicine, a smooth supply system for medicines, including narcotics, is required, particularly because there are multiple types, dosage forms, and standards for medical narcotics, and it is necessary to meet diverse needs. Therefore, collaborations on the transfer of medical narcotics centered on community cooperation pharmacies is essential; however, the transfer of narcotics is strictly regulated, and the complexity of the procedure is one of the reasons for the lack of growth in notifications [10]. The establishment of a system that reduces the economic burden, based on the deregulation of laws on distribution regarding the return of drugs to wholesalers and the transfer of medical narcotics, is important for promoting palliative medicine.

A comprehension of palliative pharmacotherapy and the proper use of medical narcotics was assessed by calculating the 2Top Ratio, which was 80.5% for ‘Appropriate use of rescue doses and patient guidance’, 70.7% for ‘Pain assessment using various scales (numerical assessment of pain such as NRS)’, 66.9% for ‘Appropriate suggestions for opioid switching’, and 60.2% for ‘WHO method of cancer pain treatment’. The results seem to show that most pharmacists had a good understanding of the appropriate use of rescue doses and patient guidance, but fewer were knowledgeable about opioid switching and the WHO method of cancer pain treatment.

To evaluate these factors, six items—“years of experience in a pharmacy”, “experience in other professions”, “type of pharmacy accreditation”, “experience in dealing with end-of-life palliative care patients with cancer who are highly dependent on medical care”, “aseptic preparation experience”, and “dispensing experience of medical narcotics by dosage form”—were cross-tabulated with four items indicating the level of understanding. In addition, the 2Top Ratio was calculated to examine the level of understanding by category. As a result, this study clarified the level of understanding of home palliative care among pharmacists at community cooperation pharmacies.

No significant differences were observed in each of the categories assessed by years of experience in a pharmacy. The levels of proficiency and understanding of palliative medicine in pharmacies were identified in previous studies as major factors affecting experience of dispensing narcotics at home [3] and in the development and enhancement of learning support systems [4], and this questionnaire inferred that levels of proficiency and understanding were not affected by years of experience in pharmacies. Pharmacists with hospital experience had a significantly higher level of understanding across categories in the areas of “pain assessment using various scales” and “appropriate suggestions for opioid switching”. Furthermore, continuous experience in teaching about cancer pain palliative medicine patients in a hospital setting has developed into subsequent assessments and prescribing suggestions, working to their advantage in terms of knowledge levels.

In terms of the type of pharmacy accreditation, pharmacies that are simultaneously accredited as health support pharmacies and specialized medical institution cooperation pharmacies have more diverse training opportunities as well as more opportunities to participate in various training courses, including hospital training to obtain accreditation, than pharmacies that are only accredited as community cooperation pharmacies, which is considered to result in a higher level of understanding.

The questionnaire on understanding of palliative drug therapy based on experience dispensing narcotics revealed that pharmacists with experience dispensing sublingual/buccal tablets and injectables had a higher level of understanding of palliative medicine in each of the assessment items than those who had dispensed only oral, patch, and suppository medications. Therefore, dispensing sublingual and buccal tablets and injectable preparations used in rescue medicine requires self-education and training in the techniques of instruction and adjustment, resulting in a higher level of understanding.

In terms of understanding palliative pharmacotherapy based on aseptic preparation, pharmacists with aseptic preparation experience showed a higher level of understanding than those without aseptic preparation experience. In addition, pharmacists who had experience in dealing with end-of-life palliative care patients with cancer and high medical dependency, including high-calorie infusions and injectable narcotics for medical use, had a higher level of understanding of palliative care than those who did not have such experience. Some physicians in charge of home care often express difficulty in dealing with end-of-life palliative care patients with cancer and high medical dependency and in administering narcotic injections [11], making it important for pharmacies to provide information and a support system [12]. Pharmacy pharmacists who already practice multidisciplinary collaboration and support systems with regard to administering narcotic injections and dealing with end-of-life palliative care patients with cancer and high medical dependency have a high level of understanding of palliative care.

Regarding aseptic facilities and the preparation of injectable infusions in pharmacies, the use of PCA pumps for pain control with injectable opioids in home palliative care was recently shown to be useful and is expected to become more widespread [12,13]. However, concerning PCA pumps in the home, the penetration rate is low and they are only used during a limited period at the end of life; therefore, even home doctors and home nurses who have experienced numerous end-of-life care cases have few opportunities to come into contact with these pumps and report that it is not a familiar tool [12].

The results obtained also showed that 52.6% of pharmacists have a good under-standing of PCA pumps, whereas 47.4% do not. Imada et al. [13] reported a high need for the use of PCA pumps in palliative medicine at home, and that pharmacy pharmacists need to provide support in this context. Therefore, pharmacists at community cooperation pharmacies need to obtain a more detailed understanding of PCA pumps, provide support for narcotic injection drug administration, including the design of administration routes, and promote awareness by holding training sessions that include lectures and exercises in collaboration with home physicians, visiting nurses, pump manufacturers, and rental companies.

Responses to the questionnaire revealed that 28.6% of respondents had experience in dispensing injectable narcotics for medical use, which was lower than oral (96.2%) and patch formulations (92.9%). The number of pharmacies with aseptic dispensing facilities was 23.3%, and together with pharmacies that do not have aseptic dispensing facilities, but have a system in place for shared use, the total was 78.9%. While many pharmacies have established a system that allows for the shared use of aseptic preparation facilities according to the criteria for certification as a community cooperation pharmacy, only 24.8% of pharmacies had aseptic preparation results. The high cost of installing and maintaining aseptic dispensing equipment makes it difficult for all pharmacies to have aseptic dispensing facilities. However, it is important to have an aseptic preparation system in place to meet the health care needs of patients and the community. In areas where demand is low, some clinics are unaware of what the capabilities of pharmacies are, and, thus, the visualization of functionally differentiated pharmacies is necessary.

Therefore, we consider it important to provide pharmacists in community cooperation pharmacies with the challenge to create learning opportunities, such as training and clinical practice, for injection preparation, cancer pain treatment methods, and appropriate prescribing suggestions, which are poorly understood. The acquisition of knowledge and skills will enable us to propose prescriptions to physicians who have difficulties in maintaining and managing the continuous administration of narcotics and injections in home care, and subsequently establish a support system. This will improve the quality of palliative care in the community, which will increase the number of requests for palliative care for terminal-stage patients with cancer who are highly dependent on medical care.

## 5. Conclusions

The present study conducted a questionnaire for pharmacists working in community cooperation pharmacies. The results obtained provide insights into the extent to which pharmacists in community cooperation pharmacies practice or understand home palliative medicine, which is one of the criteria for accreditation as a community cooperation pharmacy. It is clear that not all community cooperation pharmacies are capable of providing palliative care, including narcotic injectables, and it is important to differentiate the functions of pharmacies that provide palliative care among community cooperation pharmacies and make their capabilities visible. Few pharmacies currently accept home palliative care. The role of pharmacists in home palliative care will become more important in the future, and we aim to improve the quality of pharmacists in the community by establishing a support system.

## Figures and Tables

**Table 1 pharmacy-12-00110-t001:** Items surveyed in the questionnaire.

Item	Question No.	Questions and Answers
I. About the facility and pharmacist in charge		
	1	Age of respondents 1. 20s 2. 30s 3. 40s 4. 50s 5. 60s 6. 70s and older
	2	Years of experience as a pharmacy pharmacist (total years of experience)
	3	Please tell us about the pharmaceutical-related work you have performed. (multiple answers allowed) 1. Pharmacy pharmacist 2. Hospital pharmacist 3. Medical Representative 4. Administrative pharmacist 5. other
	4	Type of work 1. Pharmacy owner 2. Director of a corporation operating a pharmacy 3. Managing pharmacist or pharmacy director 4. Staff pharmacist
	5	Number of pharmacists enrolled in the pharmacy
	6	What type of accreditation does the pharmacy you work for have (multiple answers)? 1. Health support pharmacy 2. Community collaborative pharmacy 3. Specialized medical institution collaborative pharmacy 4. Not applicable to 1–3
II. Current status of home medical care and drug–drug collaborations		
	7	Experience in dealing with end-of-life palliative care patients with cancer who are highly dependent on medical care (e.g., patients receiving high-calorie infusions and narcotic injection drug administration) 1. Experienced 2. No experience
	8	Participation (including via the Web) in joint guidance at the time of discharge (pre-discharge conference) held at the hospital. 1. Experienced 2. No experience
	9	Achievement of holding conferences with other professions regarding team care at home (e.g., discussions with care managers and visiting nurses at meetings for persons in charge of services at patients’ homes, or visits by nurses to pharmacies to discuss medication) 1. Experienced 2. No experience
III. Sterile facilities and preparation of injectable infusions in pharmacies		
	10	Sterile preparation facilities at the pharmacy where you work. 1. Aseptic dispensing facilities (clean bench, safety cabinet) 2. There is no aseptic preparation facility, but a system is in place for shared use 3. Do not plan to handle aseptic preparation but will handle it if necessary. 4. No plans to handle them
	11	Aseptic Preparation Results 1. Have prepared aseptic preparations at own facility 2. Have experience through shared use 3. Have no experience 4. Do not know
	12	In recent years, pain control using injectable opioids with a PCA pump has become popular in-home palliative care. We would like to ask you about PCA pumps. 1. Know much about it 2. Know some about it 3. Have heard about it 4. Do not know
	13	What types of PCA pumps do you know (multiple answers allowed) 1. Disposable type (e.g., Balloonjector) 2. Mechanical type (e.g., CADD series) 3. Hybrid type (e.g., COOPDECH Amy) 4. Others
	14	System for owning PCA pumps at your pharmacy and renting them to medical institutions. 1. Experienced 2. No experience
IV. Handling of Narcotics		
	15	Track record of dispensing medical narcotics 1. Experienced 2. No experience
	16	Forms of narcotics you have dispensed (more than one answer possible) 1. Oral 2. Patch 3. Suppository 4. Sublingual/Buccal tablet 5. Injection
	17	Transfer of narcotics between pharmacies 1. Have obtained a permit for the inter-retailer transfer of narcotics with other pharmacies and have made transfers. 2. Have obtained a transfer license but have no transfer record. 3. Have not obtained but will obtain if necessary. 4. Have no plans to obtain 5. Do not know
V. Appropriate use of medical narcotics and palliative drug therapy		
	18	Assessment of pain using various scales (numerical assessment of pain, such as NRS) 1. Know in detail 2. Know to some extent 3. Have heard of it 4. Do not know
	19	About the WHO method of cancer pain treatment 1. Know in detail 2. Know to some extent 3. Have heard of it 4. Do not know
	20	Proper use of rescue doses and patient guidance 1. Know in detail 2. Know to some extent 3. Have heard of it 4. Do not know
	21	Appropriate suggestions for opioid switching 1. Know in detail 2. Know to some extent 3. Have heard of it 4. Do not know

**Table 2 pharmacy-12-00110-t002:** Attributes of survey respondents.

Item		n	(%)
Age	20s	12	4.5
	30s	80	30.1
	40s	80	30.1
	50s	66	24.8
	60s and older	28	10.5
Pharmacy Pharmacist Years of experience	1–5 years	30	11.3
	6–10 years	53	19.9
	11–15 years	54	20.3
	16–20 years	52	19.5
	21+ years	77	29.0
Pharmacy Pharmacist Work Experience	Pharmacy Pharmacist Only	160	60.2
	Hospital Pharmacist	84	31.6
	Medical Representatives	23	8.6
	Other *	17	6.4

* Pharmaceutical company research and development, CRO, government pharmacists, and drug stores.

**Table 3 pharmacy-12-00110-t003:** Pre-discharge conference participation in dealing with end-of-life palliative care patients with cancer who are highly dependent on medical care.

		Pre-Discharge Conference Participation (%)	
		Experienced (n = 116)	No Experience (n = 150)
Experience in dealing with end-of-life palliative care patients with cancer who are highly dependent on medical care (%)	Experienced (n = 130)	64.6	35.4
	No experience (n = 136)	23.5	76.5

**Table 4 pharmacy-12-00110-t004:** Sterile preparation facilities at the pharmacy where you work.

Facilities	n	(%)
Aseptic dispensing facilities (clean bench, safety cabinet)	62	24.1
There is no aseptic preparation facility, but a system is in place for shared use	148	56.5
Do not plan to handle aseptic preparation, but will handle it if necessary	15	6.2
No plans to handle it	33	12.9
Do not know	8	0.3

**Table 5 pharmacy-12-00110-t005:** Forms of narcotics you have dispensed (more than one answer possible).

Dosage Form	n	(%)
Oral	256	96.2
Patch	247	92.9
Suppository	181	68.0
Sublingual/Buccal tablet	160	60.2
Injectable	76	28.6

**Table 6 pharmacy-12-00110-t006:** Aggregate results of pharmacy pharmacists’ understanding of palliative pharmacotherapy based on years of experience and 2Top Ratio.

Question No.	Question	Category	2Top		2Under		n	2Top Ratio	*p*-Value
		Pharmacy Pharmacist Years of Experience	Know in Detail	Know to Some Extent	Have Heard of It	Do Not Know			
Q18.	Assessment of pain using various scales (numerical assessment of pain, such as NRS)	1–5 years	5	16	9	0	30	70.0	
		6–10 years	7	28	17	1	53	66.0	
		11–15 years	8	36	8	2	54	96.3	0.211
		16–20 years	2	30	18	2	52	61.5	
		21+ years	10	46	18	3	77	72.7	
		Total	32	156	70	8	266	70.7	
Q19.	About the WHO method of cancer pain treatment	1–5 years	3	15	12	0	30	60.0	
		6–10 years	7	25	17	4	53	41.5	
		11–15 years	7	28	15	4	54	64.8	0.715
		16–20 years	1	26	14	11	52	51.9	
		21+ years	10	38	19	10	77	62.3	
		Total	28	132	77	29	266	60.2	
Q20.	Proper use of rescue doses and patient guidance	1–5 years	5	22	3	0	30	90.0	
		6–10 years	6	35	12	0	53	77.3	
		11–15 years	9	35	9	1	54	81.5	0.374
		16–20 years	1	37	10	4	52	73.1	
		21+ years	14	50	9	4	77	83.1	
		Total	35	179	43	9	266	80.5	
Q21.	Appropriate suggestions for opioid switching	1–5 years	4	16	8	2	30	66.7	
		6–10 years	3	30	19	1	53	62.3	
		11–15 years	6	31	13	4	54	68.5	0.910
		16–20 years	0	34	12	6	52	65.4	
		21+ years	6	48	16	7	77	70.1	
		Total	19	159	68	20	266	66.9	

Pearson’s chi-square test.

**Table 7 pharmacy-12-00110-t007:** Aggregate results of pharmacy pharmacists’ understanding of palliative pharmacotherapy in terms of their experience in other professions and 2Top Ratio.

Question No.	Question	Category	2Top		2Under		n	2Top Ratio	*p*-Value
		Pharmacy Pharmacist Work Experience	Know in Detail	Know to Some Extent	Have Heard of It	Do Not Know			
Q18.	Assessment of pain using various scales (numerical assessment of pain, such as NRS)	Pharmacy Pharmacist Only	14	89	50	7	160	64.4	0.020 (ρ = −0.187) *
		Experienced hospital pharmacist	14	54	16	0	84	81.0	
		Other *	4	13	4	1	22	77.3	
		Total	32	156	70	8	266	70.7	
Q19.	About the WHO method of cancer pain treatment	Pharmacy Pharmacist Only	13	75	55	17	160	55.0	0.097
		Experienced hospital pharmacist	12	46	15	11	84	69.0	
		Other *	3	11	7	1	22	63.6	
		Total	28	132	77	29	266	60.2	
Q20.	Proper use of rescue doses and patient guidance	Pharmacy Pharmacist Only	17	108	29	6	160	78.1	0.338
		Experienced hospital pharmacist	14	58	9	3	84	85.7	
		Other *	4	13	5	0	22	77.3	
		Total	35	179	43	9	266	80.5	
Q21.	Appropriate suggestions for opioid switching	Pharmacy Pharmacist Only	9	89	50	12	160	61.3	0.023 (ρ = −0.187) *
		Experienced hospital pharmacist	7	59	12	6	84	78.6	
		Other *	3	11	6	2	22	63.6	
		Total	19	159	68	20	266	66.9	

* Pharmacy pharmacists with experience other than that of hospital pharmacists, such as medical representatives, pharmaceutical company development, CROs, and government pharmacists. Pearson’s chi-square test (ρ = _) *: Spearman’s rank correlation coefficient.

**Table 8 pharmacy-12-00110-t008:** Aggregate results and 2Top Ratio of understanding of palliative pharmacotherapy based on the type of pharmacy accreditation.

Question No.	Question	Category	2Top		2Under		n	2Top Ratio	*p*-Value
		Type of Pharmacy Accreditation	Know in Detail	Know to Some Extent	Have Heard of It	Do Not Know			
Q18.	Assessment of pain using various scales (numerical assessment of pain, such as NRS)	A	12	75	43	5	135	64.4	0.014 (ρ = −0.184) *
		B	15	74	27	3	119	74.8	
		C	5	7	0	0	12	100.0	
		Total	32	156	70	8	266	70.7	
Q19.	About the WHO method of cancer pain treatment	A	8	66	49	12	135	54.8	0.030 (ρ = −0.133) *
		B	16	59	27	17	119	63.0	
		C	4	7	1	0	12	91.7	
		Total	28	132	77	29	266	60.2	
Q20.	Proper use of rescue doses and patient guidance	A	16	84	28	7	135	74.1	0.014 (ρ = −0.170) *
		B	14	88	15	2	119	85.7	
		C	5	7	0	0	12	100.0	
		Total	35	179	43	9	266	80.5	
Q21.	Appropriate suggestions for opioid switching	A	8	67	51	9	135	55.6	< 0.001 (ρ = −0.229) *
		B	8	83	17	11	119	76.5	
		C	3	9	0	0	12	100.0	
		Total	19	159	68	20	266	66.9	

A: Only community cooperation pharmacies are accredited. B: Pharmacies with two accreditations: community cooperation pharmacies and health support pharmacies. C: Pharmacies that have been accredited as a specialized medical institution cooperation pharmacy in addition to community cooperation pharmacies or health support pharmacies. Pearson’s chi-square test. (ρ = _) *: Spearman’s rank correlation coefficient.

**Table 9 pharmacy-12-00110-t009:** Aggregate results and 2Top Ratio of understanding of palliative pharmacotherapy based on experience in dealing with end-of-life palliative care patients with cancer who are highly dependent on medical care.

Question No.	Question	Category	2Top		2Under		n	2Top Ratio	*p*-Value
		Experience of Teaching Medically Dependent Patients	Know in Detail	Know to Some Extent	Have Heard of It	Do Not Know			
Q18.	Assessment of pain using various scales (numerical assessment of pain, such as NRS)	Experienced	18	93	15	4	130	85.4	<0.001
		No experience	14	63	55	4	136	56.6	
		Total	32	156	70	8	266	70.7	
Q19.	About the WHO method of cancer pain treatment	Experienced	17	74	30	9	130	70.0	0.002
		No experience	11	58	47	20	136	50.7	
		Total	28	132	77	29	266	60.2	
Q20.	Proper use of rescue doses and patient guidance	Experienced	20	98	10	2	130	90.8	<0.001
		No experience	15	81	33	7	136	70.6	
		Total	35	179	43	9	266	80.5	
Q21.	Appropriate suggestions for opioid switching	Experienced	12	93	23	2	130	80.8	<0.001
		No experience	7	66	45	18	136	53.7	
		Total	19	159	68	20	266	66.9	

Fisher’s exact probability test.

**Table 10 pharmacy-12-00110-t010:** Aggregate results and 2Top Ratio of understanding of palliative drug therapy based on aseptic preparation experience.

Question No.	Question	Category	2Top		2Under		n	2Top Ratio	*p*-Value
		Aseptic Preparation Experience	Know in Detail	Know to Some Extent	Have Heard of It	Do Not Know			
Q18.	Assessment of pain using various scales (numerical assessment of pain, such as NRS)	Experienced	12	50	4	0	66	93.9	<0.001
		No experience	20	106	66	8	200	63.0	
		Total	32	156	70	8	266	70.7	
Q19.	About the WHO method of cancer pain treatment	Experienced	11	38	15	2	66	74.2	0.009
		No experience	17	94	62	27	200	55.5	
		Total	28	132	77	29	266	60.2	
Q20.	Proper use of rescue doses and patient guidance	Experienced	10	54	2	0	66	97.8	<0.001
		No experience	25	125	41	9	200	75.0	
		Total	35	179	43	9	266	80.5	
Q21.	Appropriate suggestions for opioid switching	Experienced	8	50	8	0	66	87.9	<0.001
		No experience	11	109	60	20	200	60.0	
		Total	19	159	68	20	266	66.9	

Fisher’s exact probability test.

**Table 11 pharmacy-12-00110-t011:** Comprehension of palliative pharmacotherapy based on dispensing experience of medical narcotics by dosage form and 2Top Ratio.

Question No.	Question	Category	2Top		2Under		n	2Top Ratio	*p*-Value
		Performance of Medical Narcotics	Know in Detail	Know to Some Extent	Have Heard of It	Do Not Know			
Q18.	Assessment of pain using various scales (numerical assessment of pain, such as NRS)	A	0	3	3	1	7	42.9	<0.001 (ρ = −0.398) *
		B	2	26	27	2	57	49.1	
		C	3	15	11	3	32	56.3	
		D	12	54	26	2	94	70.2	
		E	15	58	3	0	76	96.1	
		Total	32	156	70	8	266	70.7	
Q19.	About the WHO method of cancer pain treatment	A	0	3	1	3	7	42.9	<0.001 (ρ = −0.368) *
		B	2	19	29	7	57	36.8	
		C	2	13	9	8	32	46.9	
		D	9	48	27	10	94	60.6	
		E	15	49	11	1	76	84.2	
		Total	28	132	77	29	266	60.2	
Q20.	Proper use of rescue doses and patient guidance	A	0	3	2	2	7	42.9	<0.001 (ρ = −0.417) *
		B	0	33	21	3	57	57.9	
		C	4	16	9	3	32	62.5	
		D	15	70	8	1	94	90.4	
		E	16	57	3	0	76	96.1	
		Total	35	179	43	9	266	80.5	
Q21.	Appropriate suggestions for opioid switching	A	0	2	2	3	7	28.6	<0.001 (ρ = −0.440) *
		B	0	25	21	11	57	43.9	
		C	2	12	15	3	32	43.8	
		D	6	63	22	3	94	73.4	
		E	11	57	8	0	76	89.5	
		Total	19	159	68	20	266	66.9	

A: No results on medical narcotic dispensing. B: Only results on medical narcotics dispensed as oral or patch formulations. C: Only results on medical narcotics dispensed as oral or patch formulations and suppositories. D: Dispensing results on oral or patch formulations, suppositories, and sublingual/buccal products only. Dispensing results on oral or patch formulations and sublingual/buccal products only. E: Dispensing of injectable preparations; Pearson’s chi-square test. (ρ = _) *: Spearman’s rank correlation coefficient.

## Data Availability

Data are available from the corresponding author upon reasonable request.

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
