# Peer review of "Questionnaire on the Current Status and Awareness of Palliative Medicine by Community Cooperation Pharmacies"

_pharmacy, 2024, doi:10.3390/pharmacy12040110_

Round 1

Reviewer 1 Report

Comments and Suggestions for Authors

Thank you for the opportunity to review your manuscript concerning community pharmacists and their opinions regarding palliative care medicine.  It was fascinating to read your findings.  Over all, this is well presented and an easy read.  I have a few specific comments and several editorial comments that will be presented here or under the discussion of English.

Please look at lines 65-66:  I do not understand who "Pharmacy Pharmacists" are.  This phrase is chosen again on lines 69-70, without clarity.  Please tell me which pharmacies and which pharmacists.  If "Pharmacy Pharmacists" is a legal title, then please define it.

Page 3, immediately following line 93 - I believe this is Figure 1, but I am unable to find the title as a header

Palliative care involves more than managing pain.  I did not see any commentary concerning anxiety, depression, etc.  If your intent is to focus only on pain management as a part of palliative care, it would be helpful for that to be addressed in the introduction.

Comments on the Quality of English Language

The quality of language use is excellent.  My comments are offered for editorial and readability purposes.  Use those that are of interest to you and ignore the rest.

Read your manuscript from line 57 through line 62.   You've chosen to use the word "therefore" twice in a very short space.  Pick 1 of them and simply omit the other.

The word "however" is grossly overused in American English.  No sentence should ever start with that word.  Quality sentences don't need it and become stronger statements when a leading "however" is removed.  Consider line 299, line 313, and line 327.  The word however is used correctly on line 332 as it is embedded in the sentence rather than initiating it.

Reviewer 2 Report

Comments and Suggestions for Authors

This manuscript studies different aspects of the role of community pharmacist in handling of palliative medicines. 

I consider it very interesting, as it is not a role that usually play the community pharmacists but that can be of great value. To have information about the expand of this service and the knowledge and abilities that the pharmacists need to have can be of help for the implementation in other countries.

The work is well written and the methodology used by the authors gives information about the abilities and competences needed. 

In order to understand a little better, the work of the pharmacist in this area, I consider that the author should explain briefly how works the narcotic supply in the Japanese pharmacies.

Comments on the Quality of English Language

In general the English language is correct, but I consider that it could be improved. 

Reviewer 3 Report

Comments and Suggestions for Authors

REVIEW REPORT FOR THE STUDY “QUESTIONNAIRE ON THE CURRENT STATUS AND AWARENESS OF PALLIATIVE MEDICINE BY COMMUNITY COOPERATION PHARMACIES”

Journal: Pharmacy

The paper "Questionnaire on the Current Status and Awareness of Pallia-2 tive Medicine by Community Cooperation Pharmacies", performs a study based on a survey on access to healthcare for patients requiring palliative medicines at home was conducted to identify the problems of access to palliative medicines among older pharmacists working in community pharmacies.

Title and summary. The title and abstract express well the object of study, objectives, and results of the article.

Structure of the article. The contents are well organized and they adhere to the IMRaD structure. It includes a theoretical framework of the research problem and focusing on the opportunity of the study, it must be said that it is useful work due to the impact of issues related to cancer and palliative care.

Materials and methods.

Regarding the material and methods section, the methodology is tailored to the object of study and the objectives and is explained in a transparent manner while it has been validly applied to guarantee the results. However, it would be interesting to give in Material and methods section, a description on the Validation process: Once the final draft has been designed, i.e., once the information has been the information has been delimited, the questions have been formulated, defined the number of questions to be included in the questionnaire and the questionnaire and the questions have been ordered, it is time to carry out the pilot test and the evaluation of the metric properties of the scale.

- Pilot test or cognitive pre-test: Normally, the draft questionnaire is passed to 30-50 people, and it is advisable that they resemble the individuals in the individuals in the sample.

- Reliability:

The degree to which an instrument measures accurately, without error. It indicates the condition of the instrument to be reliable, i.e. capable of giving reliable and consistent results in repeated use and consistent results under similar measurement conditions.

The reliability of a measuring instrument is assessed by means of consistency, temporal stability and inter-observer agreement.

- Consistency: This refers to the level at which the different items or questions of a scale are related to each other. This homogeneity between items indicates the degree of agreement between items and the degree of agreement between items.

the degree of agreement between them and, therefore, that will determine whether they can be aggregated to give an overall score. Consistency can be checked using different statistical methods. Cronbach's alpha coefficient is a widely used statistical method. Its values range from 0 to 1. Good internal consistency is considered to exist when the alpha value is greater than 0.7.

- Temporal stability: This is the concordance obtained between the results of the test when the same sample is evaluated by the same assessor in two different situations (test-retest reliability).

The reliability (usually calculated with the intraclass correlation coefficient [ICC], for continuous variables and temporally distant evaluations) indicates that the result of the measure is stable over time. A correlation of 70% would indicate acceptable reliability.

- Inter-observer agreement. In the analysis of the level of agreement obtained when the same sample is assessed under the same sample is assessed under the same conditions by two different or at different times, the same results are obtained (inter-observer reliability). Inter-observer agreement can be analysed by means of the by means of the percentage of agreement and the Kappa index.

Validity

The degree to which a measuring instrument measures what it actually what it actually intends to measure or serves the purpose for which it was constructed. Although different types of validity are described, validity, however, is a unitary process and it is precisely it is precisely validity that will enable the correct inferences and interpretations to be correct inferences and interpretations of the scores obtained when applying a test and to establish the relationship with the establish the relationship with the construct/variable being measured.

- Content validity. This refers to whether the questionnaire, and therefore the items chosen, are indicative of the indicators of what it is intended to measure. This involves submitting the questionnaire to the assessment of researchers and experts, who must judge the questionnaire's ability to assess all the dimensions we wish to measure.

Therefore, there is no room for any calculation, only the qualitative assessments that expert researchers must make.

- Construct validity. This assesses the degree to which the instrument reflects the theory of the phenomenon or concept it measures. Construct validity ensures that the measures resulting from the responses to the questionnaire can be considered and used as a measure of the phenomenon we want to measure. It can be calculated by various methods, but the most frequent are factor analysis and the multitrait-multimethod matrix.

- Criterion validity. Relation of each subject's score to a

to a Gold Standard that is guaranteed to measure what we want to measure what we want to measure. Reference indicators are not always available, often, in practice, we resort to the use of instruments that have been endorsed by other studies or research and offer us guarantees of measure what we want to measure. Depending on the type of variables, we will use Pearson's correlation coefficients (quantitative variables) or quantitative variables) or calculation of sensitivity and specificity (qualitative variables).

Results.

The results are significant and they are presented in an adequate and understandable way not only through narration but also with self-explained tables that are also well elaborated in terms of presentation. The results justify and relate to the objectives and methods and the results are of sufficient interest with the exception noted above.

Discussion.

The discussion appropriately compares the study results with other works, highlighting the main study findings. The 38.46% of the bibliography cited in the study belongs to the previous five years.

Overall, it is an interesting study and should be considered for publication in Pharmacy, once the revisions proposed have been resolved.

Reviewer 4 Report

Comments and Suggestions for Authors

Pharmacists can play an important role in palliative care and providing services to assist patients at home.  Understanding the level of preparedness of pharmacists is important and can help guide the development of training. From this point of view, the aim of this paper is important. However, the proposed publication in its current form, is difficult to understand. It may contain data that can provide understanding in this area. However, further explanation and clarification of the study and results are required.

Abstract:

 The sentences containing “actual acceptance of palliative medicine at home has not yet been examined. In the present study a questionnaire on the acceptance system for patients requiring palliative medicine”-unsure what is meant by "acceptance"? Given that the questionnaire asks about aseptic facilities, dispensing of opioids, and knowledge of pharmacists, would it be more appropriate to refer to this as how well prepared are pharmacies and pharmacists working in community cooperation pharmacies to address the requirements for palliative medicine service?  

Introduction:

The introduction needs a clear explanation of the pharmacy system for international readers to help them understand the results obtained. The discussion section seems to explain the system somewhat. Is it correct that a system consisting of community cooperation pharmacies and specialised medical institution cooperation pharmacies was established in 2021? The discussion section then states that community cooperation pharmacies play a central role in palliative medicine at home. Are these the only types of pharmacy with this role?

From section 3 results, It would also seem pharmacies can be accredited for different functions and that community cooperation pharmacies could also be accredited as health support pharmacies and specialised medical institution cooperation pharmacies. 

The research results by Hasegawa are currently in the text as an excerpt directly from the publication in quotation marks. This needs to be summarised in the authors' own words and related to what is being proposed in the authors’ questionnaires. What did the Hasegawa study conclude- how does it differ from the current study being reported, or how does the present study add to the findings of the Hasegawa?

The paragraph commencing at line 47 (lines 47-52)  should be rewritten to be something along the lines of: The Ministry of Health, Labor and Welfare Ordinance (2021) requires community cooperative pharmacies as systems to comply with the following standards: consideration of the privacy of users, facilitate consultation, share users’ drug information with other medical institutions, have a dispensing and sales system that ensures a stable supply of medicines and a dispensing and guidance system at home? (not sure what this phrase means).

Line 61-64. Again, the word acceptance is used. Perhaps it would be better to state “Therefore, the present study conducted a questionnaire for managing pharmacists engaged in community cooperation pharmacies to investigate the actual status (or preparedness) of the system to provide appropriate service to patients requiring home palliative medicine given that this is an accreditation criterion.

Line 65 onwards: Studies by Myotoku, Harikae and Hidaka are mentioned. There is mention of pharmacy pharmacists- how do these pharmacists differ from other pharmacists? Were these pharmacists who worked in community cooperation pharmacists or other facilities? Were the surveys used in the 3 studies mentioned in line 65 the source of the questions used in this questionnaire? How did the pharmacists surveyed in the 3 mentioned studies differ from those in this study?

Section 2: Material and Methods

Section 2.3 There is reference to Top-2%. What does this mean?, it needs to be explained fully as it is used in results tables 6-10. Without knowing what this means, the reader cannot interpret results.

3.0 Results

It is noted that 266 facilities responded. Of these, it appears that 135 pharmacies were accredited as community cooperation pharmacies. Since the title of the paper is Current Status and Awareness of Palliative Medicine by Community Cooperation Pharmacies, shouldn’t the data analysis be only on these 135 participants and not 266? It would make sense then to look at the results for Tables 2 onwards using this subset of pharmacists.

The issues raised above need to be addressed before the remainder of this paper can be appropriately reviewed.

Comments on the Quality of English Language

There are some terms used in the paper, such as acceptance, which may not be the correct word to use.

Round 2

Reviewer 3 Report

Comments and Suggestions for Authors

RE-REVIEW REPORT FOR THE STUDY “QUESTIONNAIRE ON THE CURRENT STATUS AND AWARENESS OF PALLIATIVE MEDICINE BY COMMUNITY COOPERATION PHARMACIES”

Journal: Pharmacy

Authors of the paper "Questionnaire on the Current Status and Awareness of Palliative Medicine by Community Cooperation Pharmacies", have improved the work presented but, in my opinion, the need to incorporate construct validity of the questionnaire (e.g. by factor analysis) and criterion validity by analysis of Pearson's correlation coefficients remains unanswered. With these two additions, the article would be suitable for publication in the Journal Pharmacy.

Results.                  

The results are significant and they are presented in an adequate and understandable way not only through narration but also with self-explained tables that are also well elaborated in terms of presentation. The results justify and relate to the objectives and methods and the results are of sufficient interest with the exception noted above.

Discussion.

The discussion appropriately compares the study results with other works, highlighting the main study findings. The 38.46% of the bibliography cited in the study belongs to the previous five years.

Overall, it is an interesting study and should be considered for publication in Pharmacy, once the revisions proposed have been resolved.

Reviewer 4 Report

Comments and Suggestions for Authors

Introduction:

Line 33: suggest replacing live their lives with “remain at home”

 Table 2- unsure what the rationale is for performing Pearson Chi-Square test on this data (see further comment under discussion section)

3.4 Guidance and handling

Line 201: It is not evident from the data how it can be concluded that 76 pharmacists had more experience in dispensing injectable drugs than other dosage forms. Would it be more appropriate to state that 76 pharmacists had experience with dispensing injectable drugs?

Discussion

Further work is required in the discussion section to improve flow and clarity.

For some comments below, some changes to wording are suggested to assist the authors BUT authors need to determine if these align with their study observations.

Line 317: suggest using word supply rather than “reception” and sentence ended at this point. Then start a new sentence: “In this study, a survey was conducted to investigate the actual status of community cooperation pharmacies that provide services for patients requiring palliative medicine”.

Line 322-323; unsure what the significance or rationale was for performing Pearsons Chi square test (table 2) on this data; wouldn’t reporting the percentages of experienced and no experience pharmacists with pre-discharge conference participation be sufficient?

Line 325: The following sentence is suggested rather than the sentence commencing after 14.8%: “The results of our survey suggest that the percentage of pharmacists at community collaborative pharmacies providing end-of-life palliative care who participate in pre-discharge conferences is higher than previously reported”.

Line 327-330. Suggest. “It was observed that over 50% of community collaborative pharmacies who participated in the survey were not involved with dealing with end-of-life palliative care patients with cancer and high medical dependency. While the reasons for this were not a focus of this survey, another study? (8) reported reasons for not providing this service were “no requests from physicians” and lack of demand. It may also be related to lack of understanding….(9)” 

Line 338, where aseptic facilities are mentioned, could be a better place to move lines 419 to 429. The aseptic dispensing discussion could be followed by the discussion of PCA pumps (both results from this study and from the literature). Alternatively, the section on PCA pumps could be moved to be discussed in the paragraphs around line 419.

For clarity, it is suggested that the sentence commencing at line 366 be ended after the phrase “WHO method for cancer pain treatment.” The results could then be discussed. For example, the results seem to show that most pharmacists had a good understanding of the appropriate use of rescue doses and patient guidance, but fewer were knowledgeable about opioid switching and the WHO method of cancer pain treatment.

Lines 373 to 379 should be rewritten to express the examination of understanding by category more clearly.

Conclusion: 

This is a well-constructed paragraph and provides a good conclusion to the publication

Comments on the Quality of English Language

Some further work is required 

Round 3

Reviewer 4 Report

Comments and Suggestions for Authors

This paper is now much easier to read and understand the research outcomes. It is suggested that line 3 of the abstract be changed to "the actual status of the system to provide palliative medicine."

Check data in lines 457-458. It states 52.6% of pharmacists have a good understanding of PCA pumps whereas approximately  50% do not. It may be better to consider changing 50% to 47.4% (so it adds up to 100%)

Comments on the Quality of English Language

The English quality has improved significantly.
